# Transformers meet Neural Algorithmic Reasoners

## Abstract

Transformers have revolutionized machine learning with their simple yet effective architecture. Pre-training Transformers on massive text datasets from the Internet has led to unmatched generalization for natural language understanding (NLU) tasks. However, such language models remain fragile when tasked with algorithmic forms of reasoning, where computations must be precise and robust. To address this limitation, we propose a novel approach that combines the Transformer's language understanding with the robustness of graph neural network (GNN)-based neural algorithmic reasoners (NARs). Such NARs proved effective as generic solvers for algorithmic tasks, when specified in graph form. To make their embeddings accessible to a Transformer, we propose a hybrid architecture with a two-phase training procedure, allowing the tokens in the language model to cross-attend to the node embeddings from the NAR. We evaluate our resulting TransNAR model on CLRS-Text, the text-based version of the CLRS-30 benchmark, and demonstrate significant gains over Transformer-only models for algorithmic reasoning, both in and out of distribution. Finally, we empirically show that Transformer-only models distilled from TransNAR models also exhibit improved out-of-distribution generalization capabilities.

## 1 Introduction

Recent work motivated (Dudzik & Veličković, 2022) and showcased (Ibarz et al., 2022; Bevilacqua et al., 2023) the effectiveness of graph neural networks (Veličković, 2023, GNNs) at robustly solving algorithmic tasks of various input sizes, both in and out of distribution—such systems are often referred to as *neural algorithmic reasoners* (Veličković & Blundell, 2021, NARs). Provided appropriate inductive biases are used, NARs are capable of holding perfect generalisation even on $6\times$ larger inputs than ones seen in the training set, for highly complex algorithmic tasks with long rollouts (Jürß et al., 2023). NARs are, however, still relatively *narrow* forms of AI, as they require rigidly structured formatting of inputs, and they hence cannot be directly applied to problems posed in more noisy forms—such as in *natural language*—even when the underlying problem is still algorithmic in nature.

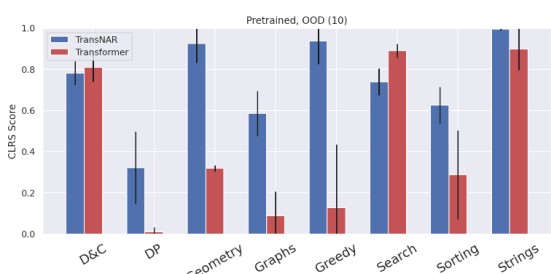

Figure 1. Our TransNAR architecture, with its direct synergy of Transformers and Neural Algorithmic Reasoners, yields clear improvements in out-of-distribution reasoning across wide categories of algorithmic tasks in CLRS-Text (Markeeva et al., 2024). Here, the $x$-axis indicates one of the eight algorithmic families of CLRS, and the $y$-axis spans the average execution accuracy across a dataset of out-of-distribution examples. TransNAR enables *emerging capabilities* in the particular out-of-distribution regime depicted here, with over 20% absolute improvement in several classes.

Conversely, the current undisputed state-of-the-art approach for modelling noisy text data are Transformer-based (Vaswani et al., 2017) language models (Anil et al., 2023; Achiam et al., 2023). In spite of their unrivalled natural language understanding properties (Wei et al., 2022), they are

also notoriously brittle when faced with even the simplest algorithmic tasks (Dziri et al., 2023)—especially if out-of-distribution generalisation is required (Anil et al., 2022).

It appears that *uniting Transformers with NARs* can lead to fruitful returns on both sides. In this paper, we explore this interface for the first time, building the **TransNAR** model.

**Contributions.** Our exploration proved fruitful. The key takeaways in this work are as follows:

1. We introduce TransNAR, a hybrid architecture combining language understanding of a Transformer with the robustness of reasoning of a pre-trained GNN-based NAR. The Transformer uses the NAR as a *high-dimensional tool* that will modulate its token embeddings.

2. We show, through an evaluation on CLRS-Text (Markeeva et al., 2024), the text-based version of the CLRS-30 benchmark, that such an NAR-augmented large language model (LLM) exhibits improved and more robust reasoning capabilities out-of-distribution (Figure 1).

3. We show that Transformer-only models distilled from TransNAR models are significantly better at out-of-distribution generalization.

Our work presents one of the most comprehensive size generalisation challenges given to Transformers to date, and the introduction of NARs moves the needle significantly.

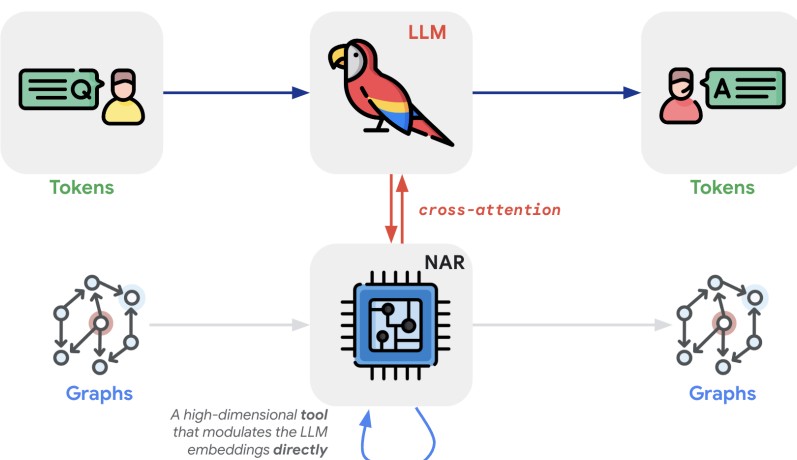

Figure 2. **Augmenting LLMs with algorithmic reasoning: a bird's eye view of TransNAR.** A large language model (LLM) consumes input tokens and produces output tokens, as common for a unimodal Transformer. The neural algorithmic reasoner (NAR) module is a graph neural network (GNN) pre-trained to execute various algorithmic computation on a collection of graph-based inputs (Ibarz et al., 2022)—the pre-training pipeline is denoted by faded arrows. Throughout its forward pass, the Transformer may access the embeddings computed by the NAR, by leveraging cross-attention (trained by learnable "glue" weights).

## 2 RELATED WORK

Our work sits at the intersection of several areas: neural algorithmic reasoning, length generalisation in language models, tool use, and multimodality. Here, we briefly survey various relevant works in each area. Due to the diversity of perspectives, to preserve brevity, we do not offer a comprehensive review of related work, but rather aim to provide an indication of specific works that inspired ours the most.

**Neural algorithmic reasoning**  NAR is, in general terms, the art of building neural networks that are capable of capturing algorithmic computation. Such capabilities can be amplified by careful choices in algorithmic alignment (Xu et al., 2020), step-wise training (Veličković et al., 2019) or contrastive objectives (Bevilacqua et al., 2023).

Recently, it was demonstrated that: (1) it is possible to learn an NAR capable of executing *multiple* algorithms simultaneously in its latent space (Xhonneux et al., 2021)—with the Triplet-GMPNN (Ibarz et al., 2022) skillfully doing so for a collection of thirty algorithms across the CLRS benchmark (Veličković et al., 2022); (2) Once trained, such NARs can be usefully deployed in various downstream tasks: reinforcement learning (Deac et al., 2021; He et al., 2022), self-supervised learning (Veličković et al., 2022), combinatorial optimisation (Georgiev et al., 2023a; Qian et al., 2023), computational biology (Georgiev et al., 2023b) and neuroscience (Numeroso et al., 2023).

Our work's use of NAR is mostly motivated by two of the works listed before: we use a relatively small, pre-trained, multi-task NAR (Ibarz et al., 2022), and deploy it in a far more scaled environment: as shown by Numeroso et al. (2023), NAR should in principle be scalable to systems that are orders-of-magnitude greater than the NAR's training distribution ($180,000\times$ in that particular case).

**Length generalisation in LLMs**  While NARs can often strongly generalise to far greater test inputs (Jürß et al., 2023), LLMs have seen significantly less success in such scenarios. We attribute this to their autoregressive, causally-masked objective, which may not always correspond to the most logical order in which outputs of algorithms should be predicted. Just as a simple example, performance of various LLMs on multiplication can be significantly improved by predicting the result in reverse order (Lee et al., 2023). Of course, on more complicated algorithms, it may be much harder to determine the best way to permute the input, and it may not be the most human-readable. Further, it was recently shown (Barbero et al., 2024) that perfectly solving certain types of problems (such as copying and counting) is fundamentally out of reach of decoder-only Transformers, due to their auto-regressive nature. Using an NAR allows the Transformer access to embeddings which have been obtained without autoregression, ameliorating this issue in part.

Knowledge of the above issues has led to a significant amount of effort being invested in building Transformers that can generalise in length. While length generalisation is not the only kind of distribution shift of interest to OOD reasoning, it is among the most easy such shifts to simulate. Accordingly, various works have attempted to induce length generalisation in LLMs, through the use of careful prompting (Zhou et al., 2022; Shen et al., 2023), randomised positional encoding (Ruoss et al., 2023), curricula (Abbe et al., 2023) or scratchpads (Anil et al., 2022). We firmly believe that an important trait of reasoning is robustness with respect to prompt quality—so long as the prompt unambiguously specifies the problem—and hence deliberately do not explore prompt modification approaches here; only randomised positions (Ruoss et al., 2023) are leveraged out of the works above in our model.

**Tool use and multimodality**  Another way to obtain robust generalisation performance is to leverage a hard-coded algorithm (also known as a *tool*) by teaching an LLM to invoke its API (Schick et al., 2023). Arguably, most of the major successes of reasoning with LLMs (Leblond et al., 2023; Romera-Paredes et al., 2023; Trinh et al., 2024) can primarily be attributed to an LLM's clever usage of a tool rather than the LLM itself, as a tool will by definition not have issues in generalising to diverse inputs.

Since our aim is to directly evaluate reasoning capabilities of LLMs, we explicitly do not permit tool use in our baselines. That being said, we envision the pre-trained NAR as a *modulator* for the Transformer's embeddings which is more robust to OOD noise. Hence, we may observe the NAR as an *"internal tool"*: rather than using raw tokens, the Transformer and NAR can communicate using their embeddings, breaking the associated algorithmic bottlenecks (Deac et al., 2021; Ong, 2023).

How to actually realise this communication and embedding exchange? For this, we turn to *multimodal LLMs* (Jaegle et al., 2021) for inspiration, since we need to integrate signals coming from two different representations of algorithmic problems (text and graph). Specifically, our exchange operator is directly inspired by vision language models (VLMs) and the cross-attention operator used in Flamingo (Alayrac et al., 2022), which offered a principled way of fusing information from text and image modalities. Similar cross-attentive operators have been used to combine GNN and Transformer representations (Song et al., 2019; Wang et al., 2020). We offer, however, the first such approach to combine them in the context of (algorithmic) reasoning and out-of-distribution generalisation, which is a setting that is particularly harmful for decoder-only Transformers.

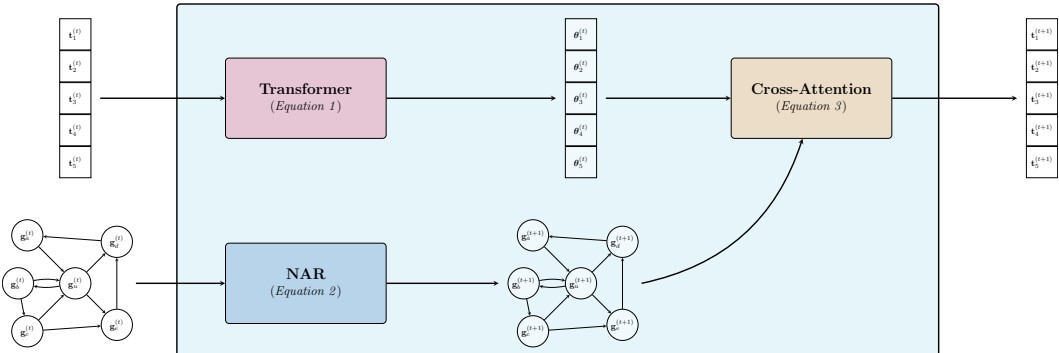

Figure 3. **TransNAR hybrid architecture.** Similar to Alayrac et al. (2022), we interleave existing Transformer layers with gated cross-attention layers which enable information to flow from the NAR to the Transformer. We generate queries from tokens while we obtain keys and values from nodes and edges of the graph. The node and edge embeddings are obtained by running the NAR on the graph version of the reasoning task to be solved. When experimenting with pre-trained Transformers, we initially close the cross-attention gate, in order to fully preserve the language model's internal knowledge at the beginning of training.

## 3 TransNAR: Augmenting Transformers with a pre-trained GNN-based NAR

This section describes our hybrid TransNAR architecture (refer to Figure 3). TransNAR accepts a dual input consisting of a textual algorithmic problem specification (of $T$ tokens) and its corresponding CLRS-30-specific graph representation (of $N$ nodes) and outputs a textual response to the problem. We can assume that, once encoded, the textual input is stored in $\mathbf{T} \in \mathbb{R}^{T \times k}$, and the graph input is stored in $\mathbf{G} \in \mathbb{R}^{N \times l}$. Note that, for simplifying the equations to follow, we make an assumption that all of the information relevant to the graph version of the problem is stored in the nodes—which is often not true in CLRS-30 (there may be edge- and graph-level inputs as well) but it doesn't change the underlying dataflow presented below.

The forward pass of TransNAR unfolds as follows. First, we properly initialise the inputs by setting $\mathbf{T}^{(0)} = \mathbf{T}$ and $\mathbf{G}^{(0)} = \mathbf{G}$. Next, to compute the representation of a step $(t+1)$, the text (token) representations are fed to the current layer of the Transformer (Vaswani et al., 2017):

$$\mathbf{\Theta}^{(t+1)} = \text{FFN}\left(\text{softmax}\left(\frac{(\mathbf{T}^{(t)}\mathbf{Q}_t)^\top \mathbf{T}^{(t)}\mathbf{K}_t}{\sqrt{d_k}}\right)\mathbf{T}^{(t)}\mathbf{V}_t\right) \tag{1}$$

where $\mathbf{Q}_t, \mathbf{K}_t \in \mathbb{R}^{k \times d_k}, \mathbf{V}_t \in \mathbb{R}^{k \times k}$ are the query, key and value transformations, respectively, and FFN is a feedforward network. In a similar manner, the graph representations are fed to the NAR layer, implementing e.g. a standard max-MPNN (Veličković et al., 2019):

$$\mathbf{g}_u^{(t+1)} = \phi\left(\mathbf{g}_u^{(t)}, \max_{1 \leq v \leq N} \psi\left(\mathbf{g}_u^{(t)}, \mathbf{g}_v^{(t)}\right)\right) \tag{2}$$

where $\psi, \phi : \mathbb{R}^k \times \mathbb{R}^k \to \mathbb{R}^k$ are learnable *message* and *update* functions, respectively, and $\max$ is the elementwise-max aggregation. Note that Equation 2 only provides pairwise interactions between nodes for brevity—in reality, our NAR is a Triplet-GMPNN (Ibarz et al., 2022), which also contains triplet interactions and a gating mechanism. Further, note that there is no timestep index on the learnable parts of the NAR—at each step, a *shared* function is applied. This aligns well with the iterative, repeated nature of algorithmic computation on graphs.

Once both streams have prepared their representations, $\mathbf{\Theta}^{(t+1)}$ and $\mathbf{G}^{(t+1)}$, the node embeddings in the graph condition the Transformer's token embeddings to produce the final outcome of the TransNAR block in the Transformer stream, inspired by Flamingo (Alayrac et al., 2022):

$$\mathbf{T}^{(t+1)} = \text{FFN}\left(\text{softmax}\left(\frac{(\mathbf{\Theta}^{(t)}\mathbf{Q}_t^\times)^\top \mathbf{G}^{(t)}\mathbf{K}_t^\times}{\sqrt{d_k}}\right)\mathbf{G}^{(t)}\mathbf{V}_t^\times\right) \tag{3}$$

where $\mathbf{Q}_t^\times, \mathbf{K}_t^\times \in \mathbb{R}^{k \times d_k}, \mathbf{V}_t^\times \in \mathbb{R}^{k \times k}$ are the key, query and value transformations of the cross-attention, respectively. No additional transformations are performed on $\mathbf{G}^{(t+1)}$ before concluding this layer.

This process repeats until the final, $N_l$-th layer, when the final text output is read out from $\mathbf{T}^{(N_l)}$. The final output is converted into token logits by a prediction head produced by the final layer, which we supervise by means of a standard next-token prediction objective.

The training of TransNAR proceeds in two phases: Firstly, prior to the start of TransNAR fine-tuning, we pre-train the NAR to robustly execute the thirty algorithms spanned by CLRS-30 (Veličković et al., 2022), in a manner similar to Ibarz et al. (2022). Such procedures are known to yield out-of-distribution generalisation at up-to-$4\times$ larger inputs in graph space. Then, the second phase (fine-tuning) can proceed. The parameters of the NAR are generally kept *frozen* during fine-tuning, as additional gradients would eliminate the model's original robustness properties. This is also, similarly, the reason why no cross-attention is performed by the graph embeddings. The LLM itself may be pre-trained over large-scale datasets (Hoffmann et al., 2022), to establish its general language priors, though we recover the same experimental findings even if the LM is randomly initialised.

## 4    EXPERIMENTS

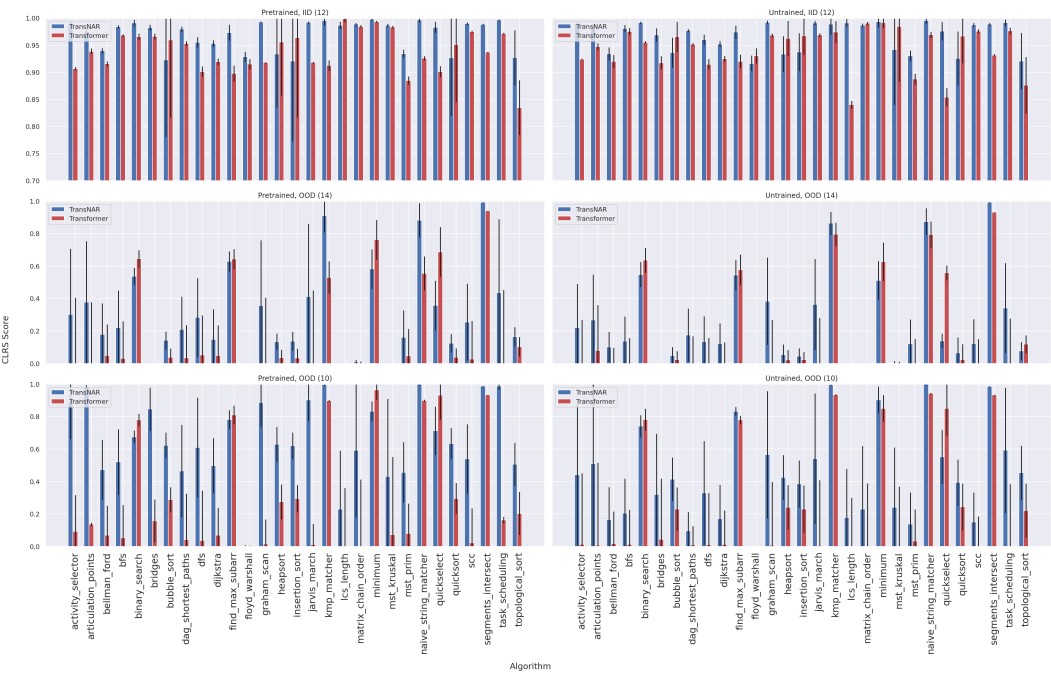

Figure 4. **TransNAR significantly outperforms the baseline Transformer.** We compare TransNAR to its corresponding Transformer baseline on various algorithms and for various input sizes: 12 is the largest size in-distribution. The other two sizes tested—10 and 14—are out-of-distribution, with the former testing interpolation and the latter extrapolation. Note that in-distribution generalisation is much easier for Transformers, and as such, we have modified the $y$-axis for this setting only to the $[0.7, 1.0]$ range. It is evident that, on most algorithmic tasks of interest, the TransNAR is capable of outperforming its baseline Transformer. Additionally, we see that this advantage is consistent across both training regimes: initial training and finetuning. The metric used is the CLRS score. Each model was trained with 4 random seeds. Error bars indicate $\pm 1$ standard deviation.

In our experimentation, we will demonstrate that the recipe offered by TransNAR admits significant benefits to out-of-distribution reasoning in language model architectures. In this section we provide details of our experimental setup.

**Transformer architecture and initialisation.** We use a decoder-only, 6 layers, transformer model from the Chinchilla family (Hoffmann et al., 2022) pretrained on MassiveText (Rae et al., 2022). In particular we use a model of 70 million parameters with a context size $2,048$. To showcase the suitability of our approach regardless of the starting point of training, we run two ablative variants. In the first, the Transformer weights are initialised with the outcome of the pre-training—emulating a *fine-tuning* scenario—and in the second, we use a fully random initialisation. In our figures and tables of results that follow, we will refer to these two setups as *"Pretrained"* and *"Untrained"*.

**Randomized positional encoding.** Previous work has emphasised the significant relevance of *randomised* positional embeddings in Transformers, especially for enabling more robust reasoning (Ruoss et al., 2023). Corresponding to previous studies on the generalization capabilities of language models, randomised positional embeddings have indeed led to significant gains on both our baselines and TransNAR, allowing more interesting reasoning behaviour to emerge in both. As such, all our experiments in this paper will use randomised positional embeddings. We provide more details in Appendix B.

**Pre-training the NAR.** Following Ibarz et al. (2022), we pre-train a multi-task MPNN-based NAR on input problem sizes of up to 16, from the CLRS-30 benchmark (Veličković et al., 2022). Owing to its graph structure formulation, such NARs are capable of significant OOD generalisation—sometimes staying competitive on graphs that are $4\times$ the size. We will attempt to utilise such models through TransNAR, to convey this rich representational knowledge into text.

**Combining cross-attention contributions from nodes and edges.** The NAR pre-trained by the method presented in Ibarz et al. (2022) produces both node and edge latent representations, and we cross-attend to both of them, as they may contain complementary useful information. To cross-attend over the edge features, $\mathbf{E}^{(t)} \in \mathbb{R}^{N \times N \times k}$, we apply Equation 3 one more time (with $\mathbf{\Theta}^{(t)}$ cross-attending over $\mathbf{E}^{(t)}$), with the caveat that we need to flatten the first and second axis of $\mathbf{E}$ into one, to make sure the dimensionalities match. We combine the cross-attention contribution from the node and edge embeddings provided by the pre-trained NAR by concatenation, followed by the application of a linear layer. We have attempted to use other reduction schemes such as summing the vectors, or applying a 2-layer MLP. We have also attempted different preprocessing schemes such as orthogonalising the contributions using the Gram-Schmidt process to ensure their algebraic complementarity before combining them. However, none of these variations have brought improvements over our original approach.

**Datasets.** We use the CLRS-Text benchmark (Markeeva et al., 2024), the text version of the CLRS-30 benchmark (Veličković et al., 2022). Note that the textual representation is directly derived from the graph-based CLRS-30 in a deterministic manner, so the two datasets convey exactly the same information. However, due to the tokenised representation, there are stringent limitations on how large of a problem size we can evaluate on without running out of context length for Chinchilla.

Accordingly, we train our algorithms on smaller problem sizes—$[4, 8]$ and 12, and evaluate on problem sizes 10 (*OOD—interpolation*), 12 (*in-distribution*), 14 (*OOD—extrapolation*).

It is worth noting that CLRS-Text is among the most challenging long-range reasoning tasks for language models, compared to the present evaluation landscape—a clear step-up in complexity from grade school math, mainly because it allows for explicitly controlling for out-of-distribution generalisation. Yet, there exists a clear polynomial-time-algorithmic description for each of them, meaning that they can be explained in relatively little parameters—certainly way less than a typical large language model of today!

The dataset comprises $10,000$ samples per algorithm per input size, making up a total of $2,400,000$ data points, split as per above into $70\%$ for training and $30\%$ for validation.

**Training details.** We train all models over seven epochs of the training data with a batch size of 256 and employ an Adam optimizer (Kingma & Ba, 2017) with a learning rate of $10^{-4}$. We apply randomized positional encoding with a maximal length of $8,192$ on top of Rotary Positional Encoding (RoPE) used in the base Chinchilla transformer (Hoffmann et al., 2022). As previously mentioned, for all TransNAR models, we keep the NAR frozen during training.

**Evaluation metrics.** We refrain from computing the accuracy of each model using exact string matching, on the grounds that this does not provide insights as to the causes of failure on a particular

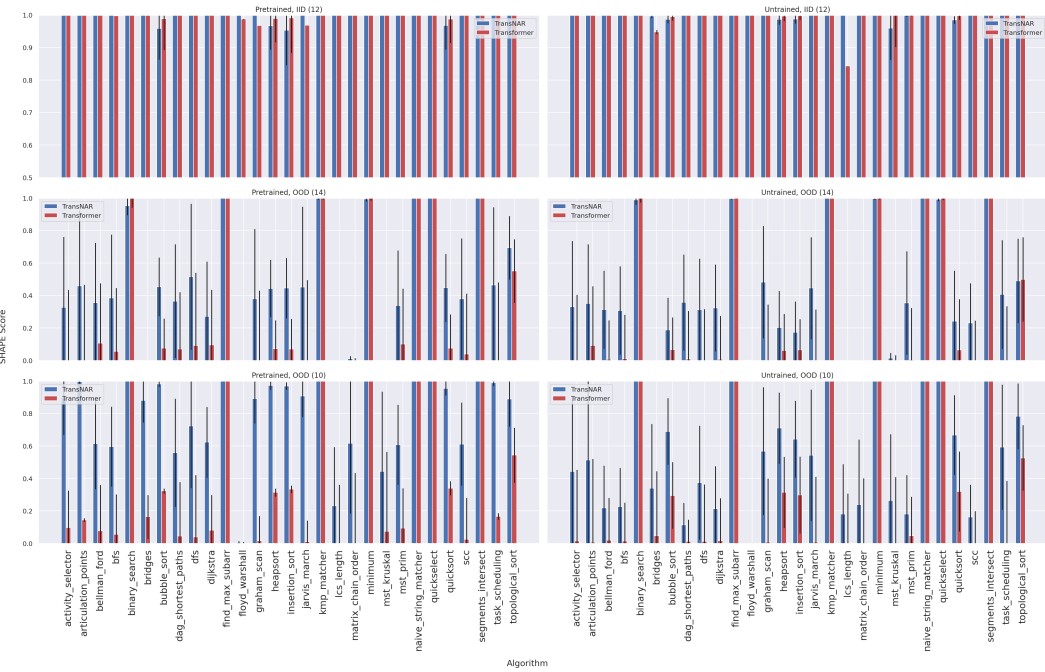

Figure 5. **Shape Score:** The TransNAR significantly outperforms its baseline in terms of producing correct shapes. This score sheds light on an obvious failure model of regular Transformers out-of-distribution: they fail to capture the seemingly trivial dependency between input size and output size, and so irrespective of the complexity of the algorithm itself. The TransNAR model manages to considerably alleviate this problem (with many emerging gains), albeit, these gains do not always lead to perfect scores, implying a fruitful direciton for future research.

datapoint, and more critically, it fails to capture how close to correctness a given model output is (as observed by Veličković et al. (2022)). Instead, we evaluate the performance of each model according to three metrics measuring capabilities of increasing complexity over the generated text:

1. The *shape score*: a binary-valued metric capturing whether the output has the right shape. For example, if we consider a sorting task, the output should have exactly the same number of elements as the input. Similarly, if the output is a matrix, we ensure its shape is consistent with both the input and the task.

2. The *parse score*: a binary-valued metric capturing whether the output is free from any illicit characters, for example, considering again a sorting task on a list of numbers, the output shouldn't contain any letters of the alphabet.

3. The *CLRS score*: The percentage of elements in the output that match the ground truth answer. This score is the one traditionally used in CLRS-30 (Veličković et al., 2022; Ibarz et al., 2022), hence its name. Note that we automatically assign a CLRS score of 0 if the shape score is 0, as there is no clear correspondence between output indices.

These multi-faceted scores are explicitly designed to capture the various failure modes of LLMs when learning to reason over text: they may overly specialise to the training problem sizes (leading to incorrect shapes at test time), fail to cope with unseen number combinations (leading to incorrect parsing), and of course, produce incorrect or inconsistent outputs, captured by the CLRS score.

### 4.1 RESULTS

We summarize our findings in Figure 4 (for CLRS score. See tabulated results in appendix A). Our results show that our TransNAR significantly outperforms the baseline Transformer overall, and on most individual algorithms, both in- and out-of-distribution. In particular, we see that our approach not only enhances existing out-of-distribution generalisation capabilities, but also causes

the emergence of these capabilities when there was a complete lack thereof—reflected in the figure by zero or near-zero performance of the baseline (Wei et al., 2022).

The analysis of shape score (Figure 5) provides an additional way to shed light on why TransNAR performed as well as it did. Recall, first, that CLRS score is necessarily zero if shapes do not match. Observing the shape scores achieved, it appears that grounding Transformer outputs in NAR embeddings significantly increases the proportion of inputs for which a Transformer will produce an output of the correct shape—indicating that this is one very specific failure mode that TransNAR helps alleviate.

We note, however, that there remain a few algorithms for which TransNAR is not able to outperform the baseline. A closer look at the results indicates that such tasks (Binary Search, Find Maximum Subarray, Minimum, and Quickselect) all involve an element of *searching* for a particular index in an input list. This hints at a unified failure mode: as these failures persist both when interpolating and extrapolating, the model as implemented is not able to generalise to novel *index boundaries* unseen in the training data. We therefore suspect that the use of *index hints*—as already demonstrated by Zhou et al. (2023)—is a promising avenue for ameliorating this behaviour. Alternatively, it might be the case that the final NAR-computed hidden states are harder to decode by the cross-attention layers in a generalisable way, and therefore might require either giving an additional capacity to the cross-attention and/or performing a more *progressive* decoding in that: instead of having all cross-attention layers decoding from the final NAR-computed hidden states, s, we could have early cross-attention layers decode from hidden states coming from earlier message passing steps, and later cross-attention layers decode from the later message passing steps.

Lastly, we provide parse scores in Appendix C—omitting them from the main text because, in most cases, parsing can be done at full accuracy.

## 4.2 DISTILLING TRANSNAR INTO A TRANSFORMER-ONLY MODEL

While our approach demonstrates favourable average performance under all out-of-distribution regimes we have evaluated, the fact that the TransNAR requires access to both textual and graph-representation limits its application to cases where a particular ground-truth executor or simulator (or prior belief about one) is available. Now that we know that TransNAR-like ideas are beneficial, we are interested in deploying such ideas into purely unimodal Transformers. Specifically, we attempt to lift the need for a second data stream by *distilling* the knowledge acquired by the trained TransNAR (*teacher*) model into a vanilla (text-only) Transformer (*student*) model.

One very interesting benefit of the distillation approach is that it allows us to train our student model on *any* problem size we want, *including* sizes that used to be considered out-of-distribution! This does not violate the desired distribution shift, as at no stage were OOD labels actually used to train any model—only the predictions of the teacher model were used as labels at those sizes. We denote this setting as "soft out-of-distribution" in what follows.

**Distillation details.** The teacher model is the TransNAR model, trained as described in previous sections. The student comprises only the Transformer model from the pipeline, and it was pre-trained on MassiveText (Rae et al., 2022).

Due to memory constraints, we focus on a proof-of-concept setting wherein the training dataset for the student comprises eight algorithms over five input problem sizes. These sizes include 4, 8 and 12—for which both ground-truth and teacher supervision are provided (in-distribution regime); and 10 and 14—for which only teacher supervision is provided ("soft OOD").

We sample $1,000$ problems per algorithm per input size, making up a total of $40,000$ training data points. For the test dataset we sample $500$ problems per algorithm for each of the out-of-distribution test input sizes 6, 10, 14 and 16, making a total of $16,000$ test data points. We train all models (students and baseline) over three epochs of the training data with a batch size of 16.

The overall loss is computed as a convex linear combination of the ground-truth next-token prediction loss (which is restricted to in-distribution problem sizes only) and the teacher distillation loss, both of which are cross-entropy losses:

$$\mathcal{L} = (1 - \alpha)\mathcal{L}|_{\text{IID}}(\boldsymbol{y}, \hat{\boldsymbol{y}}_{\boldsymbol{s}}) + \alpha\mathcal{L}(\hat{\boldsymbol{y}}_{\boldsymbol{t}}, \hat{\boldsymbol{y}}_{\boldsymbol{s}}) \qquad (4)$$

where $\alpha$ is the weight of the distillation loss; $\hat{\boldsymbol{y}}_t$ and $\hat{\boldsymbol{y}}_s$ are next-token probabilities computed by the teacher and the student respectively.

Figure 6 shows that such distilled Transformer-only models ($\alpha > 0$) are significantly better at out-of-distribution generalization than their baseline ($\alpha = 0$). This is a very encouraging result that may inform practical deployment of TransNAR-style ideas, as the distillation objective may be easily combined with any other loss within text-only Transformers.

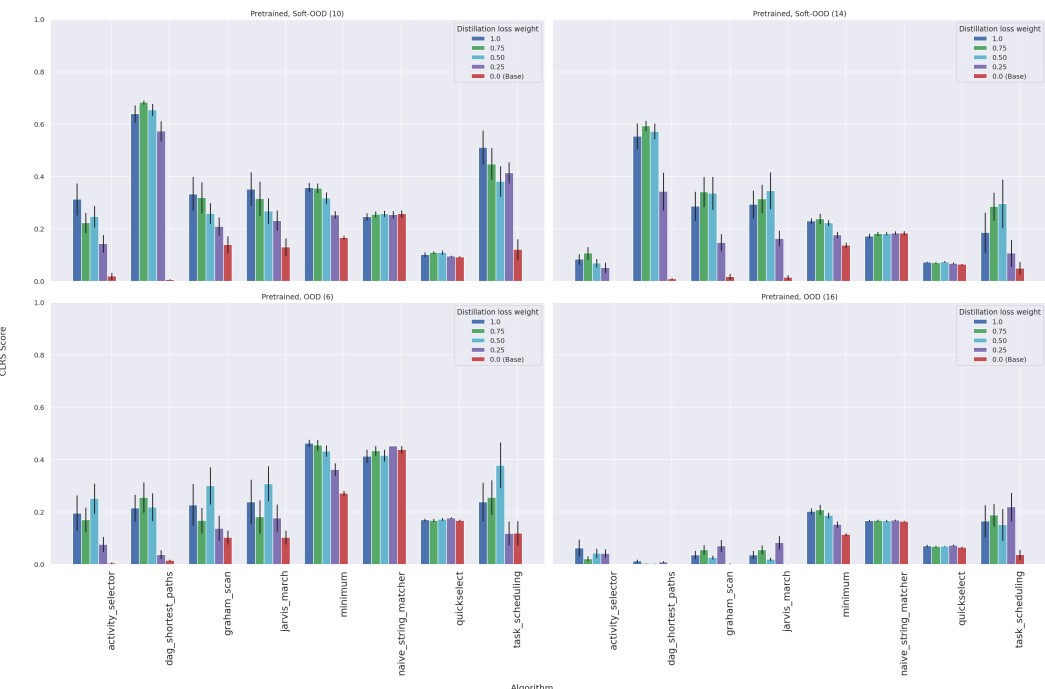

Figure 6. **TransNAR-distilled Transformer-only models significantly outperforms their baseline.** We compare TransNAR-distilled Transformer-only models to their corresponding baseline (for which distillation loss weight, $\alpha = 0$) on various algorithms and for various out-of-distribution input sizes: 6 and 10 testing interpolation, and 14 and 16 testing extrapolation. Furthermore, 10 and 14 test "soft" out-of-distribution in that problems of these sizes were seen by the student during training, but only teacher supervision was provided for them (never the ground-truth); 6 and 16 test "hard" out-of-distribution in that problems of these sizes were not seen by the student during training at all. The metric used is the CLRS score. Each model was trained with 10 random seeds. Error bars indicate $\pm 1$ standard error.

### 4.3 LIMITATIONS

While TransNAR demonstrates strong potential for enhancing out-of-distribution reasoning in language models, some key limitations warrant attention in future research:

*Dependence on Initial Graph Representation*: Although our distillation approach transfers some reasoning capabilities to a Transformer-only model, this process still relies on the initial availability of graph representations for training the teacher model. This dependence on structured data limits the applicability of TransNAR to scenarios where a clear graph representation or a reliable understanding of the underlying algorithm is present. Extending its ability (e.g. by developing domain-specialized NARs) to handle ambiguous problem specifications, commonly encountered in real-world situations, is crucial for wider practical use.

*Distillation Loss Weight Optimization*: Determining the ideal distillation loss weight ($\alpha$) appears to be task-specific and potentially sensitive to the input length. For example, a value of 0.5 seems generally good in the interpolation regime, while 0.25 seems better in the extrapolation regime. Further investigation is needed to understand how to balance ground-truth supervision and teacher distillation effectively across different scenarios. Alternatively, one might consider

using ensemble decoding techniques (such as weight-averaging (Chronopoulou et al., 2023) or majority-voting (Wang et al., 2023)), combining models trained with different values of $\alpha$ at inference-time.

## 5 CONCLUSIONS

We presented a Transformer-NAR hybrid architecture: a language model that combines the language understanding skills of a Transformer with the robust algorithmic reasoning capabilities of a pre-trained graph neural network-based neural algorithmic reasoner, to solve algorithmic tasks specified in natural language. We have demonstrated the superiority of our model over its Transformer-only counterpart on the CLRS-Text benchmark, in the in-distribution, and more importantly, in two out-of-distribution regimes, with respect to the input problem size. We have further showed that such TransNAR models can be distilled into Transformer-only models with some retention of out-of-distribution generalization capabilities.

We hope that future work will draw on our results and insights shared here, and further investigate expansions of interest, notably, datasets with more ambiguous problem specifications such as those involving mathematics, logical inference, or common sense reasoning. Developing NARs that can effectively address these more nuanced domains might require innovative approaches to graph representation, potentially moving beyond rigid structures to capture more abstract relationships and uncertainties. Nevertheless, we believe the success of TransNAR in the classical algorithm domain provides encouragement for continued investment in specialized differentiable solvers. The ability to distill such specialized models into more general-purpose language models, as demonstrated through our distillation experiments, further strengthens this argument. Such a research direction could lead to more robust and reliable reasoning capabilities in language models across a wider range of real-world applications.

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

# A TABULATED CLRS SCORE: TRANSNAR VS TRANSFORMER

## A.1 PRETRAINED, IID (12)

| algorithm | transnar_mean | transnar_std | transformer_mean | transformer_std |
|---|---|---|---|---|
| activity_selector | 0.990 | 0.003 | 0.906 | 0.036 |
| articulation_points | 0.974 | 0.005 | 0.939 | 0.010 |
| bellman_ford | 0.940 | 0.004 | 0.916 | 0.005 |
| bfs | 0.984 | 0.003 | 0.968 | 0.005 |
| binary_search | 0.991 | 0.006 | 0.966 | 0.013 |
| bridges | 0.982 | 0.005 | 0.966 | 0.007 |
| bubble_sort | 0.923 | 0.142 | 0.959 | 0.060 |
| dag_shortest_paths | 0.979 | 0.005 | 0.953 | 0.005 |
| dfs | 0.955 | 0.010 | 0.901 | 0.011 |
| dijkstra | 0.953 | 0.006 | 0.920 | 0.005 |
| find_max_subarr | 0.973 | 0.015 | 0.898 | 0.061 |
| floyd_warshall | 0.929 | 0.009 | 0.915 | 0.028 |
| graham_scan | 0.993 | 0.001 | 0.918 | 0.085 |
| heapsort | 0.934 | 0.099 | 0.956 | 0.056 |
| insertion_sort | 0.920 | 0.147 | 0.964 | 0.054 |
| jarvis_march | 0.992 | 0.002 | 0.918 | 0.083 |
| kmp_matcher | 0.996 | 0.010 | 0.912 | 0.097 |
| lcs_length | 0.987 | 0.007 | 0.999 | 0.001 |
| matrix_chain_order | 0.989 | 0.002 | 0.985 | 0.014 |
| minimum | 0.998 | 0.002 | 0.993 | 0.014 |
| mst_kruskal | 0.987 | 0.003 | 0.983 | 0.003 |
| mst_prim | 0.934 | 0.008 | 0.885 | 0.011 |
| naive_string_matcher | 0.996 | 0.004 | 0.926 | 0.068 |
| quickselect | 0.983 | 0.010 | 0.901 | 0.043 |
| quicksort | 0.926 | 0.107 | 0.951 | 0.070 |
| scc | 0.990 | 0.002 | 0.975 | 0.009 |
| segments_intersect | 0.988 | 0.001 | 0.937 | 0.009 |
| task_scheduling | 0.996 | 0.001 | 0.972 | 0.013 |
| topological_sort | 0.927 | 0.051 | 0.834 | 0.042 |

A.2    UNTRAINED, IID (12)

| algorithm | transnar_mean | transnar_std | transformer_mean | transformer_std |
|---|---|---|---|---|
| activity_selector | 0.991 | 0.001 | 0.924 | 0.010 |
| articulation_points | 0.980 | 0.006 | 0.947 | 0.006 |
| bellman_ford | 0.934 | 0.012 | 0.920 | 0.009 |
| bfs | 0.981 | 0.007 | 0.975 | 0.007 |
| binary_search | 0.992 | 0.002 | 0.955 | 0.010 |
| bridges | 0.969 | 0.012 | 0.918 | 0.125 |
| bubble_sort | 0.936 | 0.028 | 0.965 | 0.011 |
| dag_shortest_paths | 0.977 | 0.003 | 0.952 | 0.011 |
| dfs | 0.960 | 0.009 | 0.915 | 0.009 |
| dijkstra | 0.952 | 0.006 | 0.925 | 0.009 |
| find_max_subarr | 0.974 | 0.012 | 0.920 | 0.005 |
| floyd_warshall | 0.917 | 0.014 | 0.930 | 0.007 |
| graham_scan | 0.992 | 0.003 | 0.968 | 0.008 |
| heapsort | 0.934 | 0.033 | 0.962 | 0.009 |
| insertion_sort | 0.937 | 0.035 | 0.967 | 0.007 |
| jarvis_march | 0.991 | 0.004 | 0.969 | 0.007 |
| kmp_matcher | 0.989 | 0.020 | 0.974 | 0.012 |
| lcs_length | 0.991 | 0.007 | 0.840 | 0.384 |
| matrix_chain_order | 0.987 | 0.003 | 0.990 | 0.001 |
| minimum | 0.993 | 0.010 | 0.992 | 0.004 |
| mst_kruskal | 0.942 | 0.102 | 0.984 | 0.002 |
| mst_prim | 0.930 | 0.010 | 0.887 | 0.009 |
| naive_string_matcher | 0.995 | 0.005 | 0.969 | 0.009 |
| quickselect | 0.976 | 0.017 | 0.854 | 0.057 |
| quicksort | 0.926 | 0.050 | 0.967 | 0.007 |
| scc | 0.987 | 0.004 | 0.976 | 0.006 |
| segments_intersect | 0.989 | 0.002 | 0.932 | 0.010 |
| task_scheduling | 0.992 | 0.006 | 0.976 | 0.004 |
| topological_sort | 0.921 | 0.052 | 0.876 | 0.036 |

## A.3 PRETRAINED, OOD (14)

| algorithm | transnar_mean | transnar_std | transformer_mean | transformer_std |
|---|---|---|---|---|
| activity_selector | 0.302 | 0.405 | 0.000 | 0.000 |
| articulation_points | 0.377 | 0.377 | 0.000 | 0.001 |
| bellman_ford | 0.179 | 0.192 | 0.049 | 0.079 |
| bfs | 0.220 | 0.230 | 0.030 | 0.050 |
| binary_search | 0.536 | 0.053 | 0.644 | 0.051 |
| bridges | 0.000 | 0.000 | 0.000 | 0.000 |
| bubble_sort | 0.142 | 0.055 | 0.038 | 0.081 |
| dag_shortest_paths | 0.210 | 0.202 | 0.034 | 0.053 |
| dfs | 0.284 | 0.242 | 0.053 | 0.086 |
| dijkstra | 0.147 | 0.186 | 0.048 | 0.080 |
| find_max_subarr | 0.628 | 0.061 | 0.644 | 0.061 |
| floyd_warshall | 0.000 | 0.000 | 0.000 | 0.000 |
| graham_scan | 0.356 | 0.404 | 0.000 | 0.000 |
| heapsort | 0.134 | 0.050 | 0.036 | 0.078 |
| insertion_sort | 0.137 | 0.058 | 0.035 | 0.078 |
| jarvis_march | 0.412 | 0.449 | 0.000 | 0.000 |
| kmp_matcher | 0.909 | 0.100 | 0.530 | 0.194 |
| lcs_length | 0.000 | 0.000 | 0.000 | 0.000 |
| matrix_chain_order | 0.009 | 0.012 | 0.000 | 0.000 |
| minimum | 0.582 | 0.122 | 0.762 | 0.050 |
| mst_kruskal | 0.000 | 0.000 | 0.000 | 0.000 |
| mst_prim | 0.161 | 0.167 | 0.047 | 0.086 |
| naive_string_matcher | 0.882 | 0.105 | 0.555 | 0.182 |
| quickselect | 0.358 | 0.152 | 0.687 | 0.043 |
| quicksort | 0.125 | 0.057 | 0.038 | 0.085 |
| scc | 0.254 | 0.237 | 0.026 | 0.049 |
| segments_intersect | 0.992 | 0.002 | 0.940 | 0.014 |
| task_scheduling | 0.436 | 0.453 | 0.000 | 0.000 |
| topological_sort | 0.164 | 0.060 | 0.103 | 0.027 |

## A.4    UNTRAINED, OOD (14)

| algorithm | transnar_mean | transnar_std | transformer_mean | transformer_std |
|---|---|---|---|---|
| activity_selector | 0.221 | 0.270 | 0.000 | 0.000 |
| articulation_points | 0.268 | 0.280 | 0.080 | 0.196 |
| bellman_ford | 0.101 | 0.094 | 0.003 | 0.004 |
| bfs | 0.138 | 0.151 | 0.006 | 0.012 |
| binary_search | 0.548 | 0.076 | 0.635 | 0.031 |
| bridges | 0.000 | 0.000 | 0.000 | 0.000 |
| bubble_sort | 0.049 | 0.052 | 0.023 | 0.027 |
| dag_shortest_paths | 0.175 | 0.164 | 0.005 | 0.008 |
| dfs | 0.134 | 0.157 | 0.000 | 0.000 |
| dijkstra | 0.120 | 0.128 | 0.003 | 0.004 |
| find_max_subarr | 0.544 | 0.093 | 0.577 | 0.064 |
| floyd_warshall | 0.000 | 0.000 | 0.000 | 0.000 |
| graham_scan | 0.383 | 0.268 | 0.000 | 0.000 |
| heapsort | 0.055 | 0.062 | 0.022 | 0.027 |
| insertion_sort | 0.045 | 0.048 | 0.023 | 0.028 |
| jarvis_march | 0.362 | 0.280 | 0.000 | 0.000 |
| kmp_matcher | 0.863 | 0.071 | 0.796 | 0.041 |
| lcs_length | 0.000 | 0.000 | 0.000 | 0.000 |
| matrix_chain_order | 0.001 | 0.002 | 0.000 | 0.000 |
| minimum | 0.512 | 0.119 | 0.627 | 0.106 |
| mst_kruskal | 0.004 | 0.010 | 0.000 | 0.000 |
| mst_prim | 0.121 | 0.150 | 0.002 | 0.004 |
| naive_string_matcher | 0.874 | 0.082 | 0.793 | 0.043 |
| quickselect | 0.139 | 0.046 | 0.558 | 0.084 |
| quicksort | 0.064 | 0.096 | 0.023 | 0.028 |
| scc | 0.124 | 0.149 | 0.000 | 0.001 |
| segments_intersect | 0.992 | 0.002 | 0.930 | 0.008 |
| task_scheduling | 0.341 | 0.277 | 0.000 | 0.000 |
| topological_sort | 0.077 | 0.055 | 0.118 | 0.011 |

## A.5 PRETRAINED, OOD (10)

| algorithm | transnar_mean | transnar_std | transformer_mean | transformer_std |
|---|---|---|---|---|
| activity_selector | 0.887 | 0.225 | 0.092 | 0.226 |
| articulation_points | 0.983 | 0.010 | 0.137 | 0.334 |
| bellman_ford | 0.472 | 0.184 | 0.068 | 0.127 |
| bfs | 0.520 | 0.201 | 0.053 | 0.112 |
| binary_search | 0.675 | 0.039 | 0.779 | 0.049 |
| bridges | 0.845 | 0.133 | 0.158 | 0.364 |
| bubble_sort | 0.622 | 0.078 | 0.287 | 0.228 |
| dag_shortest_paths | 0.465 | 0.285 | 0.041 | 0.089 |
| dfs | 0.610 | 0.307 | 0.036 | 0.069 |
| dijkstra | 0.498 | 0.169 | 0.069 | 0.128 |
| find_max_subarr | 0.782 | 0.057 | 0.810 | 0.070 |
| floyd_warshall | 0.004 | 0.006 | 0.000 | 0.000 |
| graham_scan | 0.885 | 0.151 | 0.015 | 0.025 |
| heapsort | 0.629 | 0.107 | 0.274 | 0.217 |
| insertion_sort | 0.619 | 0.083 | 0.294 | 0.221 |
| jarvis_march | 0.903 | 0.128 | 0.011 | 0.017 |
| kmp_matcher | 0.996 | 0.005 | 0.897 | 0.102 |
| lcs_length | 0.230 | 0.359 | 0.000 | 0.000 |
| matrix_chain_order | 0.591 | 0.413 | 0.000 | 0.000 |
| minimum | 0.832 | 0.062 | 0.964 | 0.039 |
| mst_kruskal | 0.431 | 0.478 | 0.072 | 0.177 |
| mst_prim | 0.456 | 0.185 | 0.080 | 0.165 |
| naive_string_matcher | 0.997 | 0.006 | 0.898 | 0.102 |
| quickselect | 0.712 | 0.148 | 0.930 | 0.033 |
| quicksort | 0.633 | 0.095 | 0.295 | 0.203 |
| scc | 0.539 | 0.213 | 0.022 | 0.052 |
| segments_intersect | 0.987 | 0.002 | 0.933 | 0.006 |
| task_scheduling | 0.986 | 0.020 | 0.163 | 0.388 |
| topological_sort | 0.505 | 0.134 | 0.203 | 0.036 |

## A.6 Untrained, OOD (10)

| algorithm | transnar_mean | transnar_std | transformer_mean | transformer_std |
|---|---|---|---|---|
| activity_selector | 0.440 | 0.437 | 0.013 | 0.028 |
| articulation_points | 0.509 | 0.509 | 0.007 | 0.016 |
| bellman_ford | 0.165 | 0.200 | 0.016 | 0.019 |
| bfs | 0.204 | 0.213 | 0.013 | 0.025 |
| binary_search | 0.740 | 0.067 | 0.782 | 0.045 |
| bridges | 0.320 | 0.375 | 0.044 | 0.096 |
| bubble_sort | 0.414 | 0.134 | 0.230 | 0.148 |
| dag_shortest_paths | 0.096 | 0.115 | 0.010 | 0.010 |
| dfs | 0.330 | 0.318 | 0.011 | 0.012 |
| dijkstra | 0.169 | 0.210 | 0.013 | 0.018 |
| find_max_subarr | 0.832 | 0.026 | 0.779 | 0.051 |
| floyd_warshall | 0.001 | 0.001 | 0.000 | 0.000 |
| graham_scan | 0.565 | 0.391 | 0.006 | 0.012 |
| heapsort | 0.426 | 0.137 | 0.241 | 0.162 |
| insertion_sort | 0.385 | 0.145 | 0.230 | 0.153 |
| jarvis_march | 0.540 | 0.401 | 0.006 | 0.010 |
| kmp_matcher | 0.994 | 0.005 | 0.932 | 0.072 |
| lcs_length | 0.177 | 0.299 | 0.000 | 0.000 |
| matrix_chain_order | 0.230 | 0.389 | 0.000 | 0.000 |
| minimum | 0.902 | 0.082 | 0.850 | 0.082 |
| mst_kruskal | 0.241 | 0.369 | 0.000 | 0.000 |
| mst_prim | 0.138 | 0.195 | 0.033 | 0.055 |
| naive_string_matcher | 0.997 | 0.005 | 0.941 | 0.060 |
| quickselect | 0.551 | 0.166 | 0.849 | 0.065 |
| quicksort | 0.393 | 0.142 | 0.244 | 0.167 |
| scc | 0.150 | 0.183 | 0.001 | 0.001 |
| segments_intersect | 0.986 | 0.001 | 0.933 | 0.004 |
| task_scheduling | 0.592 | 0.384 | 0.001 | 0.001 |
| topological_sort | 0.453 | 0.167 | 0.220 | 0.062 |

## B   EFFECT OF RANDOMIZED POSITIONAL ENCODING

Using randomized positional encoding has benefitted both our model and the baseline. In particular, combining them with NAR hiddens led to improvements OOD, most prevalently in the interpoloation regime (at length 10), but also, to some extent, in the extrapoloation regime (at length 14). One result we found interesting, was that before instating randomized positional encoding, the OOD performance of our hybrid models was limited (in fact thresholded) by the performance of the base LLM. Concretely, if the base LLM achieved near-zero performance, the hybrid architecture would fatally share the same fate. We can see that this is no longer the case: if the base LLM uses randomized positional encoding, even if its performance is near-zero, that of the hybrid architecture can still be reasonably good. This is illustrated in the second column of the figure 4, for example on the Graham Scan, Jarvis March, MST Prim algorithms.

## C   PARSE SCORES

Please see Figure C for the parse scores of various models at various sizes.

## D   SOFT- AND HARD-OOD RESULTS OF DISTILLATION

We compare the performances across various distillation coefficients on the soft- and hard-OOD problem sizes in Figure 8. Critically, distillation almost-always significantly improves performance compared to the baseline (irrespective of distillation loss coefficient). As we drift further out-of-

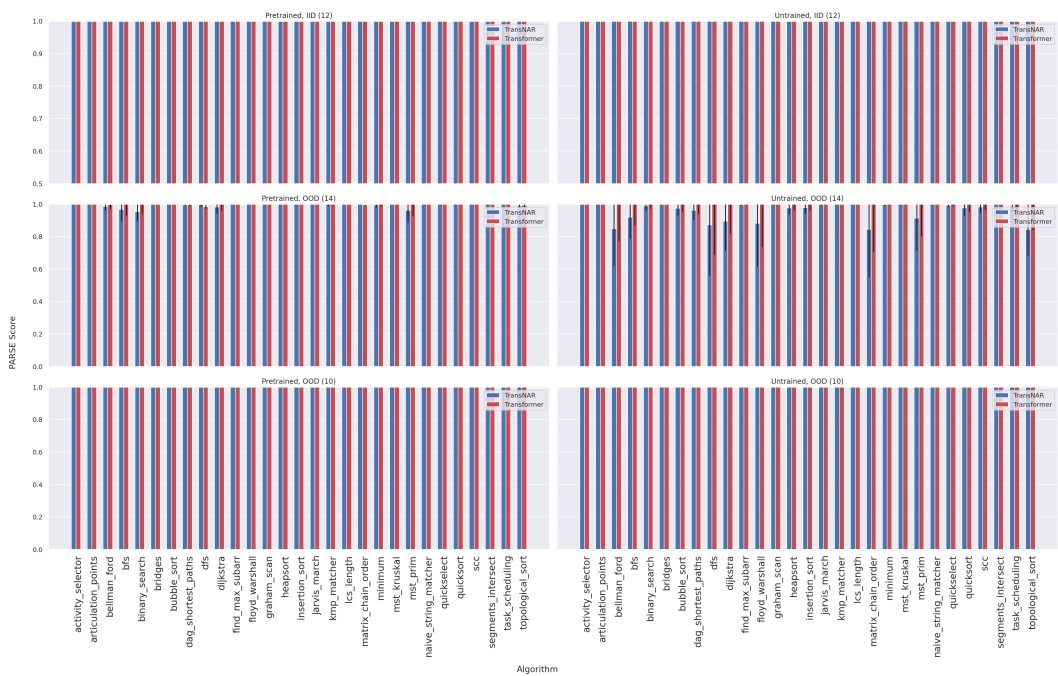

Figure 7. **Parse Score:** We can see that for a few algorithms, the TransNAR architecture falls behind the baseline in the extrapolation regime likely due to an unsufficient capacity of the cross-attention in charge of decoding from the NAR's outputs.

Length: 6

| algorithm | baseline (distill 0.0) | distill 0.5 | distill 1.0 |
|---|---|---|---|
| activity_selector | 0.005 +- 0.003 | **0.252** +- 0.057 | 0.196 +- 0.067 |
| dag_shortest_paths | 0.015 +- 0.003 | **0.219** +- 0.053 | 0.216 +- 0.051 |
| graham_scan | 0.103 +- 0.027 | **0.300** +- 0.072 | 0.227 +- 0.080 |
| jarvis_march | 0.103 +- 0.027 | **0.308** +- 0.066 | 0.238 +- 0.086 |
| minimum | 0.272 +- 0.009 | 0.433 +- 0.021 | **0.463** +- 0.014 |
| naive_string_matcher | **0.438** +- 0.014 | 0.416 +- 0.023 | 0.413 +- 0.024 |
| quickselect | 0.168 +- 0.004 | **0.172** +- 0.006 | 0.170 +- 0.006 |
| task_scheduling | 0.119 +- 0.046 | **0.378** +- 0.087 | 0.239 +- 0.073 |

Length: 10

| algorithm | baseline (distill 0.0) | distill 0.5 | distill 1.0 |
|---|---|---|---|
| activity_selector | 0.020 +- 0.014 | 0.247 +- 0.041 | **0.313** +- 0.060 |
| dag_shortest_paths | 0.007 +- 0.002 | **0.655** +- 0.023 | 0.639 +- 0.033 |
| graham_scan | 0.140 +- 0.032 | 0.259 +- 0.040 | **0.334** +- 0.066 |
| jarvis_march | 0.132 +- 0.033 | 0.268 +- 0.050 | **0.352** +- 0.065 |
| minimum | 0.167 +- 0.008 | 0.319 +- 0.022 | **0.358** +- 0.018 |
| naive_string_matcher | **0.258** +- 0.013 | **0.258** +- 0.011 | 0.247 +- 0.014 |
| quickselect | 0.093 +- 0.003 | **0.110** +- 0.008 | 0.102 +- 0.009 |
| task_scheduling | 0.121 +- 0.040 | 0.382 +- 0.058 | **0.511** +- 0.065 |

Length: 14

| algorithm | baseline (distill 0.0) | distill 0.5 | distill 1.0 |
|---|---|---|---|
| activity_selector | 0.000 +- 0.000 | 0.069 +- 0.016 | **0.085** +- 0.020 |
| dag_shortest_paths | 0.010 +- 0.003 | **0.573** +- 0.029 | 0.555 +- 0.049 |
| graham_scan | 0.017 +- 0.011 | **0.336** +- 0.063 | 0.287 +- 0.056 |
| jarvis_march | 0.015 +- 0.009 | **0.346** +- 0.071 | 0.294 +- 0.053 |
| minimum | 0.138 +- 0.011 | 0.223 +- 0.011 | **0.230** +- 0.011 |
| naive_string_matcher | **0.184** +- 0.008 | 0.182 +- 0.006 | 0.172 +- 0.009 |
| quickselect | 0.065 +- 0.003 | **0.075** +- 0.003 | 0.074 +- 0.002 |
| task_scheduling | 0.051 +- 0.024 | **0.296** +- 0.093 | 0.187 +- 0.077 |

Length: 16

| algorithm | baseline (distill 0.0) | distill 0.5 | distill 1.0 |
|---|---|---|---|
| activity_selector | 0.001 +- 0.001 | 0.043 +- 0.017 | **0.062** +- 0.032 |
| dag_shortest_paths | 0.002 +- 0.001 | 0.004 +- 0.001 | **0.013** +- 0.006 |
| graham_scan | 0.002 +- 0.002 | 0.026 +- 0.008 | **0.036** +- 0.016 |
| jarvis_march | 0.001 +- 0.001 | 0.020 +- 0.006 | **0.036** +- 0.016 |
| minimum | 0.114 +- 0.005 | 0.187 +- 0.011 | **0.203** +- 0.011 |
| naive_string_matcher | 0.164 +- 0.003 | 0.166 +- 0.003 | **0.167** +- 0.003 |
| quickselect | 0.065 +- 0.003 | 0.069 +- 0.002 | **0.071** +- 0.005 |
| task_scheduling | 0.037 +- 0.019 | 0.151 +- 0.061 | **0.165** +- 0.061 |

Figure 8. Distillation results across several soft- and hard-OOD sizes.

distribution, distillation fully on logits (1.0) outperforms partially combining distillation with next-token prediction (0.5).