# OpenReview forum: "Transformers meet Neural Algorithmic Reasoners"
_ICLR.cc/2025/Conference — Submitted to ICLR 2025_

### Official Review · Reviewer_79uu · 2024-10-30

**Soundness:** 3
**Presentation:** 3
**Contribution:** 3
**Rating:** 6
**Confidence:** 3

**Summary:**

The paper proposes a new model architecture, TransNAR, that combines a Transformer model with a Neural Algorithmic Reasoner via a cross-attention mechanism. The motivation is to leverage the reasoning and generalization ability in NAR representation to improve the performance of the transformer model. The experiment shows promising results in the out-of-distribution regime and over 20% absolute improvement in several classes.

**Strengths:**

The method is straightforward and well-motivated. Experiment results also show that it performs well on the targeted tasks compared to standard transformer architecture.

**Weaknesses:**

The proposed method is domain-specific. It's hard to imagine how this method would benefit broader domains, as the NAR requires a specific type of input. Combining this method with existing large language models would also require further experiments. Figure 5 and 6 are hard to read.

**Questions:**

1. How can we combine this method with a pretrained language model? So that we can leverage the math reasoning ability in state-of-the-art language models.
2. Have you compared with using GNN-based NAR alone?
3. How does a pretrained language model perform on these tasks?

---

> ### Author Response · Authors · 2024-11-13
> **Reply to Reviewer 79uu**
>
> Thank you for your encouraging comments and for supporting our work!
>
> > The proposed method is domain-specific. It's hard to imagine how this method would benefit broader domains, as the NAR requires a specific type of input.
>
> It is indeed the case that TransNAR as presented is not applicable to problems where graphs are not provided or derived. The motivation of our work was to show that:
>
> * The knowledge stored in NARs _can_ be meaningfully leveraged by Transformers (via TransNAR).
> * The knowledge captured within TransNAR can be _distilled_ back into a general-purpose Transformer, even though they are purely autoregressive.
>
> We believe our distillation result to be the proof-of-concept that can be used as a blueprint for empowering broader domain learning in the future!
>
> > How can we combine this method with a pretrained language model? So that we can leverage the math reasoning ability in state-of-the-art language models.
>
> The most direct application of our work here would include fine-tuning the language model using our distillation loss, potentially leveraging LoRA-style techniques for efficiency. We could also use TransNAR-style models to generate additional training data for fine-tuning.
>
> > Have you compared with using GNN-based NAR alone?
>
> We are very upfront about NARs’ performance in the early stages of our paper:
>
> > _Provided appropriate inductive biases are used, NARs are capable of holding perfect generalisation even on 6× larger inputs than ones seen in the training set, for highly complex algorithmic tasks with long rollouts (Jürß et al., 2023)._
>
> It is well-known that GNN-based NARs can generalise much better than LLMs on these tasks. However, their lack of auto-regression and causal masking would make them much harder to train at scale over general purpose text data. The aim of our paper is to see whether some of these benefits can be upstreamed into LLMs, which _can_ be trained at scale more easily and efficiently.
>
> > How does a pretrained language model perform on these tasks?
>
> We refer the Reviewer to the CLRS-Text paper by Markeeva et al., where two-shot performance for Gemini 1.5 Flash is reported. Generally speaking, pre-trained LLMs are quite poor at executing these algorithms even across small problem sizes, with sorting perhaps being the only semi-notable exception (up to length ~14).

---

### Official Review · Reviewer_mvbM · 2024-11-02

**Soundness:** 2
**Presentation:** 2
**Contribution:** 2
**Rating:** 3
**Confidence:** 4

**Summary:**

The paper presents a TransNAR neural architecture that is a hybrid of Transformer and Neural Algorithmic Reasoner (NAR) architectures. The Transformer part of the model is allowed to cross-attend to NAR states. The model requires a hybrid input: the Transformer part expects text and the NAR part consumes a graph. The paper tests the proposed architecture on CLRS-Text, which is a text version of an algorithmic benchmark called CLRS-30. The model is trained on both textual inputs and the corresponding graph. The paper’s main claims are that (1) TransNAR performs better than Transformer. (2) that TransNAR teacher can produce extra training data for the Transformer student, and that this data increases the generalization of the latter.

**Strengths:**

The paper was clear enough for me to understand the main idea. I appreciate that the paper contained a “Limitations” section.

**Weaknesses:**

The paper proposes a very straight-forward idea and presents unsurprising results. Of course a hybrid of a general-purpose model (transformer) and a task-specific model (NAR) will perform better on the specific task. Especially when the model requires special graph-structured data, which is the case for TransNAR.

The distillation results were hard for me to process and verify. The error bars (constructed on just 3 samples!) are very large. An obvious baseline is missing: distilling a NAR model to Transformer.

On the writing side, the paper could benefit from the following improvements:
- Give more context to the reader:
   - better explain CLRS-30 benchmarks
   - better explain NAR architecture
- Include a rigorous textual analysis of performance numbers (e.g." in 3 cases out of 7 our model is significantly better than the baseline (p-value = 0.01)"), in addition to presenting all numbers on all tasks in large overwhelming figures
- Include a tabular version of the results

**Questions:**

Did you look into how the distillation data that TransNAR produces is different from the groundtruth?

---

> ### Author Response · Authors · 2024-11-13
> **(First) Reply to Reviewer mvbM**
>
> We would like to thank you for commending the clarity of the paper and the simplicity of the approach, as well as proposing multiple interesting avenues for improvement!
>
> We would like to initiate the discussion early, as we are not certain we completely comprehended all your suggestions – keeping in mind that our experiments may take up to 4--5 days to complete, we want to make sure that we have understood well while there’s still plenty of time to gather additional results!
>
> Note that this is not a complete response; we will be preparing updates to the paper in parallel while we are awaiting your thoughts.
>
> > Of course a hybrid of a general-purpose model (transformer) and a task-specific model (NAR) will perform better on the specific task. Especially when the model requires special graph-structured data, which is the case for TransNAR.
>
> We are in agreement here, though we are afraid you may not be considering our message in its entirety. Indeed, we agree it should be obvious that a hybrid model should have a better performance on this task.
>
> We find it significantly less obvious that these improvements can be distilled back into a standard text-based Transformer, and this is also a key result of our work. This is because:
>
> * Distillation is a procedure originally aimed at transferring knowledge from a larger model to a smaller one, with both assumed to share the same input and output spaces. Here we adapt it to upstreaming the OOD generalization capability of a hybrid model with multiple modalities, back into a unimodal LLM.
> * We are distilling knowledge gained from a hybrid of a non-autoregressive (GNN) and autoregressive (LLM) model into a purely autoregressive (LLM) model, and it is well known that generalisation can be harmed by autoregression (see, e.g., Barbero et al. (NeurIPS’24)). The fact that distillation allows us to convey these benefits in spite of that is not obvious to us!
>
> Would you be in agreement with these points? We are very happy to discuss them further!
>
> > The distillation results were hard for me to process and verify. The error bars (constructed on just 3 samples!) are very large.
>
> We appreciate this remark, and ask: would reducing the variance through increasing the number of seeds be sufficient to address your concern?
>
> We have already kicked off these experiments; the reason why we ask if this is sufficient _now_ is because these experiments are fairly memory intensive and slow to run. They require caching logits of the teacher model, which are quite expensive given the large vocabulary size of the underlying language model (~32k).
>
> > An obvious baseline is missing: distilling a NAR model to Transformer.
>
> Could you please clarify how this baseline could be constructed?
>
> From our understanding of distillation, it is mainly applicable between two models when their output spaces match, so that the student can be trained to mimic either the outputs or the logits of the teacher.
>
> The NAR and Transformer do not have compatible output spaces, however—the Transformer produces language tokens while the NAR produces output signals on a graph.
>
> Are you suggesting that we manually convert the NAR outputs into text and then supervise the Transformer on this via next-token prediction?
>
> > Give more context to the reader (CLRS-30, NAR, tabulation and analysis)
>
> We appreciate the reviewer’s remark and commit to add a brief introduction to both benchmarks and illustrate with one sample, as well as adding a tabular version of the results and elaborating on the analysis – this will be added to a revision during the discussion period, and we will let you know once we’ve done so.
>
> > Did you look into how the distillation data that TransNAR produces is different from the groundtruth?
>
> The distillation data produced by the TransNAR are logits – not final predictions. They can only really be compared with ground-truth solutions after decoding into text. We can get a feel for how accurate the decoded data is (compared to the ground-truth) via the TransNAR performance metrics in our paper.
>
> Would you like us to do any particular qualitative study on top of this?

---

> > ### Comment · Reviewer_mvbM · 2024-11-18
> > **let's zoom into distillation**
> >
> > Thanks for your response. Let's talk about the distillation experiments in more detail, as this is what you have focused your response on.
> >
> > I have two concerns there:
> > - You still need to provide Trans-NAR with structured inputs for distillation. So one can run NAR on this structured input. However, you emphasize the fact that the outputs spaces of Trans-NAR and NAR are different. It is a very particular set of assumptions that you consider - that aligned structured and textual inputs are available, yet there is no useful procedure to produce textual outputs from structured outputs to directly use NAR as the teacher. Can you please confirm my understanding here? I do not find this set of assumptions particularly realistic. In synthetic data generation one often generates naturalized versions of structured data. Through a procedure like this NAR could be the teacher of Transformer directly, without using Trans-NAR.
> > - I appreciate your readiness to improve the experimental results about distillation, and I understand the compute constraints. I will take a look at your new results if the are ready during the discussion period. Sadly, the current version of the paper puts the burden of interpreting dense plots with large error bars on the reader. The reader has to count in how many cases distillation makes no difference, helps for all values of \alpha, or helps only for a carefully chosen version of \alpha. Major revisions are needed in this part of the paper.

---

> ### Author Response · Authors · 2024-11-19
> **Tabulated 10-seed distillation results**
>
> Thank you for engaging with us on distillation experiments! We find them to be an important part of our paper and are willing to improve the presentation there.
>
> On the first point, we believe there is a slight misunderstanding. We wrote:
>
> > Are you suggesting that we manually convert the NAR outputs into text and then supervise the Transformer on this via next-token prediction?
>
> Which is exactly offering to convert NAR outputs into text & train the Transformer directly on them -- we just weren't sure we understood correctly! We will try to kick off NAR teacher experiments in the time we have left.
>
> On the other, we are delighted that our 10-seed distillation results have converged, and significantly reduced standard errors. Please see below for all soft- and hard-OOD results.
>
> Key results:
> * Distillation almost-always significantly improves performance compared to the baseline (irrespective of distillation loss coefficient).
> * As we drift further out-of-distribution, distillation fully on logits (1.0) trumps partially combining distillation with next-token prediction (0.5).
>
> Please let us know if these results (and their tabulation!) appropriately address your concerns!
>
> Length: 10
> | algorithm            |   baseline (distill 0.0) |        distill 0.5 |        distill 1.0 |
> |:---------------------|-------------------------:|-------------------:|-------------------:|
> | activity_selector    | 0.020 +- 0.014           | 0.247 +- 0.041     | **0.313** +- 0.060 |
> | dag_shortest_paths   | 0.007 +- 0.002           | **0.655** +- 0.023 | 0.639 +- 0.033     |
> | graham_scan          | 0.140 +- 0.032           | 0.259 +- 0.040     | **0.334** +- 0.066 |
> | jarvis_march         | 0.132 +- 0.033           | 0.268 +- 0.050     | **0.352** +- 0.065 |
> | minimum              | 0.167 +- 0.008           | 0.319 +- 0.022     | **0.358** +- 0.018 |
> | naive_string_matcher | **0.258** +- 0.013       | **0.258** +- 0.011 | 0.247 +- 0.014     |
> | quickselect          | 0.093 +- 0.003           | **0.110** +- 0.008 | 0.102 +- 0.009     |
> | task_scheduling      | 0.121 +- 0.040           | 0.382 +- 0.058     | **0.511** +- 0.065 |
>
>
> Length: 14
> | algorithm            |   baseline (distill 0.0) |        distill 0.5 |        distill 1.0 |
> |:---------------------|-------------------------:|-------------------:|-------------------:|
> | activity_selector    | 0.000 +- 0.000           | 0.069 +- 0.016     | **0.085** +- 0.020 |
> | dag_shortest_paths   | 0.010 +- 0.003           | **0.573** +- 0.029 | 0.555 +- 0.049     |
> | graham_scan          | 0.017 +- 0.011           | **0.336** +- 0.063 | 0.287 +- 0.056     |
> | jarvis_march         | 0.015 +- 0.009           | **0.346** +- 0.071 | 0.294 +- 0.053     |
> | minimum              | 0.138 +- 0.011           | 0.223 +- 0.011     | **0.230** +- 0.011 |
> | naive_string_matcher | **0.184** +- 0.008       | 0.182 +- 0.006     | 0.172 +- 0.009     |
> | quickselect          | 0.065 +- 0.003           | **0.075** +- 0.003 | 0.074 +- 0.002     |
> | task_scheduling      | 0.051 +- 0.024           | **0.296** +- 0.093 | 0.187 +- 0.077     |
>
>
> Length: 6
> | algorithm            |   baseline (distill 0.0) |        distill 0.5 |        distill 1.0 |
> |:---------------------|-------------------------:|-------------------:|-------------------:|
> | activity_selector    | 0.005 +- 0.003           | **0.252** +- 0.057 | 0.196 +- 0.067     |
> | dag_shortest_paths   | 0.015 +- 0.003           | **0.219** +- 0.053 | 0.216 +- 0.051     |
> | graham_scan          | 0.103 +- 0.027           | **0.300** +- 0.072 | 0.227 +- 0.080     |
> | jarvis_march         | 0.103 +- 0.027           | **0.308** +- 0.066 | 0.238 +- 0.086     |
> | minimum              | 0.272 +- 0.009           | 0.433 +- 0.021     | **0.463** +- 0.014 |
> | naive_string_matcher | **0.438** +- 0.014       | 0.416 +- 0.023     | 0.413 +- 0.024     |
> | quickselect          | 0.168 +- 0.004           | **0.172** +- 0.006 | 0.170 +- 0.006     |
> | task_scheduling      | 0.119 +- 0.046           | **0.378** +- 0.087 | 0.239 +- 0.073     |
>
>
> Length: 16
> | algorithm            |   baseline (distill 0.0) |        distill 0.5 |        distill 1.0 |
> |:---------------------|-------------------------:|-------------------:|-------------------:|
> | activity_selector    | 0.001 +- 0.001           | 0.043 +- 0.017     | **0.062** +- 0.032 |
> | dag_shortest_paths   | 0.002 +- 0.001           | 0.004 +- 0.001     | **0.013** +- 0.006 |
> | graham_scan          | 0.002 +- 0.002           | 0.026 +- 0.008     | **0.036** +- 0.016 |
> | jarvis_march         | 0.001 +- 0.001           | 0.020 +- 0.006     | **0.036** +- 0.016 |
> | minimum              | 0.114 +- 0.005           | 0.187 +- 0.011     | **0.203** +- 0.011 |
> | naive_string_matcher | 0.164 +- 0.003           | 0.166 +- 0.003     | **0.167** +- 0.003 |
> | quickselect          | 0.065 +- 0.003           | 0.069 +- 0.002     | **0.071** +- 0.005 |
> | task_scheduling      | 0.037 +- 0.019           | 0.151 +- 0.061     | **0.165** +- 0.061 |

---

> > ### Comment · Reviewer_mvbM · 2024-12-01
> > **increased "contribution" score to "fair"**
> >
> > Dear authors, thank you for adding a more statistically significant version of the results to the paper. I find that this is an improvement to the paper, but other issues remain (incremental nature of the work, testing on a single dataset, lack of data generation baseline, and so on).

---

> > > ### Author Response · Authors · 2024-12-01
> > >
> > > Dear Reviewer mvbM,
> > >
> > > Thank you for acknowledging our response and increasing your contribution score!
> > >
> > > > lack of data generation baseline
> > >
> > > We would like to remark we are still running the experiment with the new data generation baseline, and hope to have it in time to report during the remainder of the discussion period.
> > >
> > > We are hopeful that its inclusion might be seen as favourable from your side.
> > >
> > > > incremental nature of the work
> > >
> > > It appears that our initial response has not convinced you that the distillation results are sufficiently nontrivial and unexpected to be non-incremental. Is there anything additional you would like to see -- beyond the aforementioned direct NAR data generation approach, which we are still running! -- to improve your confidence in this regard?
> > >
> > > > testing on a single dataset
> > >
> > > We are somewhat confused that you would remark this at such a late stage in the discussion period.
> > >
> > > Your initial review never mentioned "testing on a single dataset" as a weakness, and this remark also somewhat neglects the fact that CLRS is not really a single dataset, but a collection of thirty separate algorithmic problems. Furthermore, these thirty problems are generally both harder _and_ broader in scope than what features in most published length generalisation papers.

---

### Official Review · Reviewer_gZZh · 2024-11-03

**Soundness:** 3
**Presentation:** 3
**Contribution:** 3
**Rating:** 6
**Confidence:** 4

**Summary:**

This paper introduces a transNAR architecture by having both texts and graphs as input and using cross-attention to combine the strengths of both. Systematic results on CLRS-text demonstrate significant improvements in a number of problems, even though no improvements are achieved in some of them.

**Strengths:**

The paper uses graphs and texts so that the new system can use the natural language inputs as well. The combination leads to significant performance improvements over the baseline systems. While the integration uses cross attention very similar to those in multimodal models, the benefits seem significant.

**Weaknesses:**

The paper is sketchy in covering important technical details. For example, the abstract states a two-phase training procedure is used; yet the main paper does not mention the two-phase training procedure even once. Related to this, the paper describes the system wide issues only from lines 213 to line 223 without mentioning some known issues. For example, the particular implementation of the transNAR uses 6 layers and it is well known beyond two to three layers oversmoothing becomes an issue in graph neural networks. The paper freezes the weights for graph embeddings, but multimodal models rely on a shared embedding space; the differences between the two embedding spaces should be a factor, yet there is no mention about that.

A major strength of the paper is the improvements as highlighted in Figure 1. However, the baselines do not reflect the actual state of the art. One naive way to combine NAR and transformers for CLRS-text is to use start of the art transformers to convert the problems into the graphic form required by NAR and then apply NAR. Also the transformer model used in the paper is far from the state of the art; as a baseline, the authors should use the performance of a fine-tuned state-of-the art transformer model on the CLRS-text dataset. In addition, it seems the input to the transformer model is only the text but the transNAR has dual inputs; the impact of the dual inputs should be factored in when stating the improvements.

**Questions:**

- Is every iteration the same as illustrated in Figure 3? It seems it is not as the final output is only from the last layer; for other layers, the outputs from both modules feed to the next layer as inputs. Could you revise Figure 3 to show the entire process accurately?

- Given that the proposed transNAR fails to improve on some of the problems, could you please comment on the limits of the NAR approach? While it can generalize over a larger range of lengths, NAR and transNAR would still lack the generalization and robustness of correct algorithms.

- Reasoning is inherently sequantial but it seems the graphs are not. Could you comment on how the graphs emulate the sequential nature of reasoning steps?

- One of the known issues with graphs is the oversmoothing issue. Could you comment how that affects the NAR and transNAR?

- While the improvements over the baseline seem impressive, could ypu give the specific contributions beyond combining NAR and transformers? It seems to me a fair comparison is to use a straightforward combination as the baseline rather than the transformers alone, whose limitations are known in this area.

- Could you provide references for transformers' natural langaue UNDERSTANDING properties? The paper mentioned that transformers have unrivaled such properties (Section 1)  and at the same time they are limited in generalization due to their autoregressive, causally-masked objective (Section 2). These two do not seem to go together well.

- Would you opens-source the code code for your transNAR and baseline models? As many of the papers are from one group due to the unfair advantages created, it is not acceptable to the research community.

- More generally, would you be committed to the principle that scientific research should be for the common good and accessible to all?

- Could you comment why you chose to call Q as Key and K as Query in equations (1) and (3)?

- How would you pronounce NAR? Please make sure it is consistent in the paper (for example in line 065 as "a NAR-" and as "an NAR" in many other places).

---

> ### Author Response · Authors · 2024-11-13
> **Reply to Reviewer gZZh (Part I)**
>
> Thank you very much for the positive and supportive review, and for the high degree of detail in your remarks!
>
> > The paper is sketchy in covering important technical details.
>
> We will address the specific points you raised below, but if there are any other points you would like us to address that aren’t already covered by our rebuttal response, please let us know!
>
> We intend to upstream any relevant changes into the paper later during the discussion period, and we will notify you when we’ve done so.
>
> > For example, the abstract states a two-phase training procedure is used; yet the main paper does not mention the two-phase training procedure even once.
>
> We apologise for not explicitly addressing the “two-phase” remark beyond the abstract, this was an oversight on our side and it will be fixed.
>
> For completeness, we remark here that the two phases are referring to
> (1) pre-training the NAR (following Ibarz et al., LoG’22), and
> (2) fine-tuning the TransNAR using the NAR’s embeddings (following our paper).
>
> > Related to this, the paper describes the system wide issues only from lines 213 to line 223 without mentioning some known issues.
>
> Thank you for remarking this, we are very happy to supplement this discussion with additional issues that may arise in training such models.
>
> > For example, the particular implementation of the transNAR uses 6 layers and it is well known beyond two to three layers oversmoothing becomes an issue in graph neural networks.
>
> The GNN layers used in NARs (including ours!) prefer the max aggregation function – a standard choice since the NEGA paper (Veličković et al., ICLR’20) – and since max leverages a sharp decision across its neighbourhood, it is not vulnerable to over-smoothing, even over very long horizons.
>
> > The paper freezes the weights for graph embeddings, but multimodal models rely on a shared embedding space; the differences between the two embedding spaces should be a factor, yet there is no mention about that.
>
> Thank you for remarking this; our approach mimics that of Flamingo, which also freezes the weights for its image embedding model. We acknowledge that this is not the only way to perform multimodality successfully, and will mention this explicitly in our discussion, along with a note about the potential challenges in differing embedding spaces.
>
> > One naive way to combine NAR and transformers for CLRS-text is to use start of the art transformers to convert the problems into the graphic form required by NAR and then apply NAR.
>
> This is indeed a very interesting approach and we will explicitly note it in our discussion!
>
> In the meantime, we prompted a few state-of-the-art models to extract feature matrices that could be used as NAR inputs from the textual data.
>
> While they were able to extract some interesting graph-style featurisations of the text data, we note that our NARs require highly bespoke data, including the detailed internal states of the algorithms considered, which no LLM was able to extract at this time. As such, we will not be able to feed such outputs into NAR.
>
> > Also the transformer model used in the paper is far from the state of the art; as a baseline, the authors should use the performance of a fine-tuned state-of-the art transformer model on the CLRS-text dataset.
>
> We acknowledge that our Chinchilla-style base model is not present state-of-the-art at such a model scale; however we do believe our comparison is fair, because the TransNAR model also uses that same Chinchilla-style model as the starting point.
>
> To address your question, we would like to point you to the CLRS-Text paper (Markeeva et al., 2024), where results for fine-tuning a modern Gemma 2B model on this data are provided.
>
> Generally it does not seem that scale or algorithmic improvements of the Gemma 2B model have made a significant difference to the models’ OOD capability compared to Chinchilla: on most algorithms, we observe that Gemma (much like Chinchilla) can learn to fit the algorithms well in-distribution, and out-of-distribution most algorithms promptly collapse.
>
> > In addition, it seems the input to the transformer model is only the text but the transNAR has dual inputs; the impact of the dual inputs should be factored in when stating the improvements.
>
> We agree with you that the duality of its inputs are providing TransNAR with an advantage and are happy to directly address this.
>
> That being said, we also wish to point you to our distillation experiments (Section 4.2.) where these capabilities are transferred back into simple Transformers, leading to improved OOD performance while no longer requiring dual inputs.
>
> _Answers to be continued in next message..._

---

> > ### Author Response · Authors · 2024-11-13
> > **Reply to Reviewer gZZh (Part II)**
> >
> > Continuing our answers from Part I:
> >
> > > Is every iteration the same as illustrated in Figure 3? It seems it is not as the final output is only from the last layer; for other layers, the outputs from both modules feed to the next layer as inputs. Could you revise Figure 3 to show the entire process accurately?
> >
> > You are correct, and we will revise Figure 3 to showcase multiple layers instead of just one!
> >
> > > Given that the proposed transNAR fails to improve on some of the problems, could you please comment on the limits of the NAR approach? While it can generalize over a larger range of lengths, NAR and transNAR would still lack the generalization and robustness of correct algorithms.
> >
> > Of course, we will gladly comment on the limits of NARs in this regard in the paper.
> >
> > While NARs do offer superior generalisation on algorithmic tasks compared to most other neural network approaches, they are still not exact algorithms, and will make mistakes provided a sufficient distribution shift is encountered at test time.
> >
> > Accordingly, we will emphasise that our aim here is to improve the algorithmic generalisation capabilities of LLMs, so that they can be applicable in the wider range of scenarios for algorithmic problems, while acknowledging that out-of-distribution generalisation is a very hard problem for neural networks in general, and we do not think of these models as a pure drop-in replacement for algorithmic computation.
> >
> > > Reasoning is inherently sequantial but it seems the graphs are not. Could you comment on how the graphs emulate the sequential nature of reasoning steps?
> >
> > The graphs themselves are not sequential (in fact they are assumed permutation-symmetric); the reasoning steps emerge through the temporal axis of repeatedly iterating a recurrent GNN layer (the NAR) over such a graph structure.
> >
> > > One of the known issues with graphs is the oversmoothing issue. Could you comment how that affects the NAR and transNAR?
> >
> > As per our reply above, we follow standard NAR practices and use the max aggregation function in our NAR—this significantly reduces vulnerability to oversmoothing due to sharp decisions over neighbourhoods.
> >
> > > While the improvements over the baseline seem impressive, could ypu give the specific contributions beyond combining NAR and transformers?
> >
> > We believe that the key additional contribution beyond combining the two models is distilling these capabilities back into simple Transformers (through our experiments in Section 4.2.).
> >
> > The successful distillation allows us to transfer some of the knowledge gathered within TransNAR back into more standard Transformers, making this knowledge accessible in models that can operate in settings where graphs are not provided.
> >
> > > Could you provide references for transformers' natural langaue UNDERSTANDING properties? The paper mentioned that transformers have unrivaled such properties (Section 1) and at the same time they are limited in generalization due to their autoregressive, causally-masked objective (Section 2). These two do not seem to go together well.
> >
> > Thank you for pointing out this dichotomy; this is something we are well-aware and are happy to explicitly provide references.
> >
> > The language understanding capabilities are unrivaled in decoder-only Transformers because they are able to leverage training at scale to make impressive inferences from relatively unstructured textual corpora—most other specialised models (including our NAR) require the data to be prepared in a very specific format before they will show meaningful returns. A standard reference for these abilities, and their emergence at scale, could be “Emergent Abilities of Large Language Models” (Wei et al., TMLR’22).
> >
> > That being said, the impressive levels of natural language-based predictions tend to only appear _within the distribution_ the decoder-only Transformer has been trained on, and tend to suffer when pushed outside of this distribution. As a simple example, in our algorithmic tasks, the Transformers perform quite impressively on the problem sizes they’ve been fine-tuned on, yet their performance quickly collapses after we add even 1–2 nodes to the problems.
> >
> > >  Would you open-source the code code for your transNAR and baseline models? As many of the papers are from one group due to the unfair advantages created, it is not acceptable to the research community.
> >
> > Yes, we commit to open-sourcing a reference implementation of TransNAR as well as the relevant baseline models.
> >
> > > More generally, would you be committed to the principle that scientific research should be for the common good and accessible to all?
> >
> > Yes, we are strongly committed to that principle and intend to uphold it.
> >
> > _Answers to be continued in the next message..._

---

> > > ### Author Response · Authors · 2024-11-13
> > > **Reply to Reviewer gZZh (Part III)**
> > >
> > > Continuing our answers from Part II:
> > >
> > > > Could you comment why you chose to call Q as Key and K as Query in equations (1) and (3)?
> > >
> > > Thanks for the remark, this was an omission on our part – we will correct it and properly name these parameters in our next revision.
> > >
> > > > How would you pronounce NAR? Please make sure it is consistent in the paper (for example in line 065 as "a NAR-" and as "an NAR" in many other places).
> > >
> > > Thank you for pointing this out! We would pronounce NAR as [en-eh-ar] (spelled-out) and therefore “an NAR” is the version we would go for. We will harmonise this across the paper in the next revision.
> > >
> > > This concludes our responses -- please let us know if there is anything you would like to discuss further! We thank you again for your supportive assessment of our work.

---

> > > > ### Comment · Reviewer_gZZh · 2024-11-25
> > > >
> > > > Thank you for answering my questions and addressing my concerns.
> > > >
> > > > I have read all the reviews and the authors' replies so far. I think my original assessment is still accurate. I think the paper is solid but the contributions are incremental.

---

### Official Review · Reviewer_6nT4 · 2024-11-14

**Soundness:** 2
**Presentation:** 2
**Contribution:** 2
**Rating:** 5
**Confidence:** 4

**Summary:**

This paper proposes combining transformers with graph neural networks (GNNs) to build enhanced neural algorithmic reasoners. The core approach involves a cross-attention module, enabling language transformers to leverage information from a pretrained GNN. Experiments on the CLRS-Text benchmark indicate that the proposed method improves both in- and out-of-distribution performance.

**Strengths:**

1. The TransNAR approach is well-motivated, and the core concept is clearly articulated.

2. The method achieves competitive performance compared to a vanilla Transformer.

3. The study explores a distilled version of the TransNAR transformer, demonstrating improved out-of-distribution robustness over a transformer trained from scratch without GNN support.

**Weaknesses:**

1. The novelty is somewhat limited, as combining text-based transformers with GNNs via cross-attention is well-explored in previous works, such as [1] and [2].
[1] https://aclanthology.org/Q19-1002.pdf. Semantic Neural Machine Translation Using AMR.
 [2] https://aclanthology.org/2020.tacl-1.2.pdf. AMR-To-Text Generation with Graph Transformer.

2.Both Figures 4, 5 and 6 are not clear enough. I cannot clearly read them in a print version. Please consider to use at least a table to show the results.

3. Only the Chinchilla base transformer is used in experiments. Testing with open-source pretrained LLMs like LLaMa and Mixtral could provide broader insights.

4. Experiments are limited to a single dataset with pre-constructed input graphs, leaving questions about generalizability. Testing on automatically constructed graph datasets, which may be less accurate but useful, could strengthen the study’s applicability.

**Questions:**

Please refer to Point 3 and Point 4 in the weaknesses section.

---

> ### Author Response · Authors · 2024-11-14
> **(First) Reply to Reviewer 6nT4**
>
> Thank you for the kind words about the motivation of our work and recognising the strength of our comparative evaluation!
>
> As we have done with Reviewer mvbM previously, we would like to initiate the discussion with you early, as we are not certain we completely comprehended all your suggestions. Some of our experiments may take up to 4–5 days to complete, and we want to make sure that we have understood well while there’s still plenty of time to gather additional results!
>
> Note that this is not a complete response; we will be preparing updates to the paper in parallel while we are awaiting your thoughts.
>
> > The novelty is somewhat limited, as combining text-based transformers with GNNs via cross-attention is well-explored in previous works, such as [1] and [2].
>
> Thank you for pointing out these references to us, and we will certainly cite and explore them in our related work discussion! We were not aware of these specific works.
>
> To be clear: we are not claiming to be the first to combine GNNs and Transformers in general, but we do claim to be the first to combine them in the context of (algorithmic) reasoning and out-of-distribution generalisation, which is a setting that is particularly harmful for decoder-only Transformers.
>
> > Both Figures 4, 5 and 6 are not clear enough. I cannot clearly read them in a print version. Please consider to use at least a table to show the results.
>
> We appreciate your concern about the figures’ readability, and commit to tabulating these results in our next revision! We will let you know once the paper has been updated.
>
> > Only the Chinchilla base transformer is used in experiments. Testing with open-source pretrained LLMs like LLaMa and Mixtral could provide broader insights.
>
> We appreciate your remark on this!
>
> While it would certainly be possible to include additional base Transformers to the mix, we are averse to it for two reasons:
>
> * It would take a significant amount of time and resources, which we would rather dedicate to other experiments at this time.
> * It is _highly unlikely_ to lead to a different conclusion to what we already observed with Chinchilla.
>
> To provide direct evidence towards the second point, we would like to point you to the results in the CLRS-Text paper (Markeeva et al., 2024), where results for fine-tuning a modern, open-source, Gemma 2B model on this data are provided.
>
> Generally, it does not seem that scale or algorithmic improvements of the open Gemma 2B model have made a significant difference to the models’ OOD capability compared to Chinchilla: on most algorithms, we observe that Gemma (much like Chinchilla) can learn to fit the algorithms well in-distribution, and out-of-distribution, performance on most algorithms promptly collapses (also like Chinchilla).
>
> As such, it is our expectation that, were we to e.g. repeat our experiments with Gemma 2B, we’d still observe significant collapse out-of-distribution (as Markeeva et al. did), and TransNAR would be able to supplement that.
>
> We have no strong reason to expect any other open language model to behave differently.
>
> > Experiments are limited to a single dataset with pre-constructed input graphs, leaving questions about generalizability. Testing on automatically constructed graph datasets, which may be less accurate but useful, could strengthen the study’s applicability.
>
> We appreciate the Reviewer’s concern!
>
> Firstly, we’d like to note that CLRS-Text should not be considered a “single dataset” – realistically it encompasses _thirty_ different algorithmic skills that are learned and evaluated simultaneously.
>
> Second, the focus of our work is on out-of-distribution generalisation – specifically, length generalisation – of language models. It is absolutely standard to leverage carefully constructed synthetic tasks (where outputs can be computed at arbitrary input lengths) for this purpose; see, e.g., “Exploring Length Generalization in Large Language Models” from Anil et al. (NeurIPS’22), which uses two crafted tasks (Parity and Variable Assignment).
>
> Taken together, the tasks in CLRS-Text are arguably more challenging than most tasks typically seen in length generalisation papers, encompassing more generic polynomial time procedures beyond simple arithmetic – spanning path-finding, geometric algorithms, string matching, sorting, minimum spanning tree finding, and more.
>
> It is because of this that we believe our collection of datasets is sufficient for the purpose we aim to show. Please let us know what you think!
>
> All that being said, we are always open to suggestions for strengthening our work’s experimental section – please, let us know if there are any specific datasets you would like us to explore, and we can attempt to pursue them in the remainder of the discussion period.

---

> > ### Comment · Reviewer_6nT4 · 2024-12-03
> > **Acknowledgement**
> >
> > Thank you for the response.
> >
> > For the dataset explanation,  I think it is reasonable and but it is better to highlight the usefulness of the CLRS-text to general readers.
> >
> > For the experiments on open-sourced language models,  I can understand the intention is to show the OOD generalization ability. However, it would be more interesting to see how much TransNAR can help an open-sourced LLM (i.e. Gemma 2B in the original CLRS-text paper).
> >
> > The novelty is still my major concern which might not be enough  for an ICLR paper.   For correcting the dataset understanding, I will improve my score to 5.

---

### Meta-Review · Area_Chair_5Ceo · 2024-12-21

**Metareview:**

The paper proposes a novel hybrid architecture called TransNAR, which combines the strengths of Transformers and Graph Neural Networks to enhance neural algorithmic reasoning. The authors introduce a two-phase training procedure where the Transformer cross-attends to node embeddings from a pre-trained GNN. The model is evaluated on the CLRS-Text benchmark, demonstrating significant improvements in algorithmic reasoning tasks, both in and out of distribution. Additionally, the paper shows that Transformers distilled from TransNAR models exhibit improved out-of-distribution generalization.


Strengths:
* Well-motivated approach, core-concept is clearly articulated (despite some missing technical details and presentation issues)
* Strong empirical results (although baseline appropriateness is questionable)
* Shows that the hybrid model can be distilled back into a transformer-only model that outperforms a text-only base transformer model

Weaknesses:
* As pointed out by multiple reviewers, the idea of combining Transformers with GNNs via cross-attention is not entirely new and has been explored in previous works.
* Limited experimental scope: no evaluation with pretrained open-source LLMs, a single (meta-)dataset considered
* Missing baselines:
* Poor presentation of results, hard-to-read bar charts with minimal difference, large error bars computed on few seeds

What's missing:
* As suggested by virtually all the reviewers, the presentation of results in this paper requires severe polishing. Font size, y-axis dynamic range, etc, large error bars due to small number of seeds, all make it hard to appreciate and assess the significance and soundness of the results. Additionally, some additional simple baselines, including those suggested by the reviewers, would have gone a long way towards contextualizing the performance of the proposed method. Finally, although the authors provided some additional results in the form of table during the rebuttal, those did not seem to have made it to the revised version of the paper. Additionally, the authors promised a "data generation baseline" in response to mvbM's review, which I could not find in the comments or revised pdf.

Overall, this paper is an interesting albeit incremental contribution that suffers from various issues that in my opinion need to be resolved before publication in a venue like ICLR.

**Additional Comments On Reviewer Discussion:**

The main points raised by the reviewers and corresponding authors' response were:
* Reviewer gZZh: Raised concerns about the clarity of the method and the baselines. The authors provided detailed responses and made some revisions, but the reviewer maintained that the contributions were incremental.
* Reviewer mvbM: Criticized the straightforward nature of the approach and the distillation results. The authors provided additional results that somewhat improved the statistical significance of the results, but the reviewer still found the work incremental and lacking broader applicability.
* Reviewer 6nT4: Highlighted the limited novelty and the need for additional experiments with different baselines. The authors addressed some points but could not fully satisfy the reviewer’s concerns.
* Reviewer 79uu: Appreciated the motivation and empirical results but noted the domain-specific nature of the method, questioned whether this would generalize to broader domains, and pointed out the issues with the figures.

The authors made efforts to address the reviewers’ concerns during the rebuttal period, including running additional experiments and improving the clarity of their presentation. However, the fundamental issues regarding the novelty and generalizability of the approach remained unresolved. Given the mixed reviews and the incremental nature of the contributions, the decision is to reject the paper. Furthermore, as pointed out above, the additional experiments do not seem to have been included in the revised version of the paper. Other issues that would have been very straightforward to resolve, like the issues with the figures, were not resolved either.

Although some of the reviewers stopped engaging with the authors despite repeated nudges, after reading their reviews and responses in details, I don't think their main concerns were addressed. I have also heavily discounted the additional results provided by the authors because: (i) they were not included in the revised paper, which is the medium we are ultimately deciding on (and the reason why ICLR allows for paper revisions); and (ii) they were only provided in response to a single author, and thus only guaranteed to have been checked by that reviewer.

---

### Decision · Program_Chairs · 2025-01-22

Reject